# Efficacy of a Psycho-Educational and Socio-Emotional Intervention Programme for Adolescents

**DOI:** 10.3390/ijerph19138153

**Published:** 2022-07-02

**Authors:** Susana Sánchez-Herrera, Eloísa Guerrero-Barona, Diana Sosa-Baltasar, Juan Manuel Moreno-Manso, Miguel Ángel Durán-Vinagre

**Affiliations:** 1Faculty of Education and Psychology, University of Extremadura, Avda. De Elvas S/N, 06006 Badajoz, Spain; ssanchez@unex.es (S.S.-H.); eloisa@unex.es (E.G.-B.); jmmanso@unex.es (J.M.M.-M.); 2Secondary School Castelar, Avda. Santiago Ramón y Cajal 2, 06001 Badajoz, Spain; dsosab@gmail.com

**Keywords:** self-concept, anxiety, socio-emotional skills, adolescents, programme effectiveness

## Abstract

The aim of this article is to evaluate the effectiveness of the implementation of a psycho-educational intervention programme. The objective of this programme was to achieve optimisation of self-concept and basic socio-emotional skills for comprehensive development in the adolescent stage. The sample consisted of 402 students from 19 groups from public secondary schools in the province of Badajoz. A quasi-experimental pretest-posttest design was applied with a control group using the Trait Meta-Mood Scale, Self-Concept Form 5, and the State-Trait Anxiety Inventory. The analysis showed that the programme was effective in the family and social dimensions of self-concept, while in the other dimensions, the changes were not statistically significant. The results for emotion perception, understanding and regulation show that there were no statistically significant differences in the experimental group, although there were significant differences in the control group.

## 1. Introduction

Educational guidance has been evolving and transforming over the last few years and has assumed a role as a necessary resource to cope with the continuous changes in education [1,2,3,4,5,6]. This raises the need for a new profile of psychology and education professionals to contribute to the personal, social, and professional development of students [7,8,9,10]. It should also be noted that the work of guidance counsellors is of vital importance, as one of their main functions is to develop and implement intervention proposals that favour the comprehensive development of students, taking their needs as a starting point [11,12,13,14].

All this has led to a deeper reflection for some years now regarding the direction being taken by the education system and psycho-educational guidance in particular, with the aim of ensuring that these skills are duly developed.

Adolescents have to go through very complicated socio-emotional situations and find themselves without the resources they need to provide effective responses to them. This is why young people need to be taught socio-emotional skills that help them to successfully face and manage these situations, given that the ability to manage emotions is not yet fully matured at this stage of life or is in the process of development [15,16,17,18].

It has been scientifically proven that emotional intelligence and socio-emotional skills are a very useful personal tool to effectively face multiple situations in everyday life [19,20,21,22,23], and especially in the educational context [24,25,26], making it necessary to foster their maximum development [27,28,29,30]. 

In this regard, numerous programmes have been designed and implemented to address the socio-emotional development of the adolescent population. Highlights in Spain include the following: “Emotional Stability” [31,32], “Socio-Emotional Development and Prevention of Violence” [33], “Emotional Intelligence” [34], “Assertive Resolution of Conflicts Between Peers” [35], “Physical-Mental Health” [36], “Socio-Emotional Factors and Psychosomatic Symptoms” [37], “Emotional Education” [38,39], and “Improving Personal Skills for Success” [40], among others.

Two of the authors of this article have heard and sought to provide answers to the demands of both students and teachers during the course of their professional careers in secondary schools as guidance counsellors. Students, both boys and girls, come to guidance counsellors with problems relating to symptoms of anxiety. They have difficulties understanding what is wrong with them, do not know how to act in certain situations, are overcome by fears, fail to control their impulses, have difficulty relaxing and a tendency to perceive situations as threatening. In addition, they frequently verbalise feelings of hatred towards the people around them or even towards themselves, an indication of self-esteem problems. Some even involved episodes of self-harm [41,42,43].

With regard to teachers, year after year, we have come across attitudes of refusal to teach in certain grades, more specifically in the second year of compulsory secondary education (ESO). Teachers complained mainly about classroom behaviour and discipline problems, which affected not only the teaching and learning process but also coexistence in the school setting.

Other authors argue that although numerous programmes have been developed in recent years relating to adolescent socio-emotional development, many of them lack a solid theoretical and scientific basis and should be viewed with great scepticism [44]. In this sense, it is not enough to simply design and apply programmes that aim to develop socio-emotional skills; it is also necessary to evaluate these interventions to obtain empirical data regarding the extent of their validity, and also to detect the aspects that could be improved [45,46,47]. Accordingly, the objective of this study is to evaluate the effectiveness of the implementation of a socio-emotional intervention programme, considering the variables of self-concept (academic, social, emotional, family, and physical), emotional intelligence (emotion perception, emotion understanding and emotional regulation) and anxiety (state anxiety and trait anxiety) of students in secondary education.

## 2. Materials and Methods

### 2.1. Sample

A total of 19 groups from different secondary schools in the province of Badajoz took part. The sample consisted of 402 students, 191 females (47.51%) and 211 males (52.49%). The age of the participants ranged between 12 and 19 years (M = 14.1; SD = 1.60). According to academic year, the sample was distributed as follows: 126 students from 1st year ESO (31.34%), 105 from 2nd year ESO (26.12%), 50 from 3rd year ESO (12.44%) and 64 students from 4th year ESO (15.92%). A total of 27 students (6.72%) from the Initial Professional Qualification Programmes (PCPI) and 30 subjects (7.46%) from the Curricular Diversification (DC) Programme also participated. A total of 223 students (55.5%) out of the total number of participants had never repeated a year, while 179 (44.5%) had at some stage repeated a year.

The students were selected using non-probability convenience sampling. Some participants were excluded from the study because they did not provide informed parental consent and others because the questionnaire was incomplete, resulting in the loss of important information.

### 2.2. Instruments

In order to assess the constructs under study, they were evaluated before and after application of the intervention programme using the following instruments with psychometric guarantees of reliability and validity.

Self-Concept Form 5 (AF5; [48]) was applied collectively. This assesses several dimensions of self-concept. Previous research has confirmed the adequate psychometric properties of this instrument [49] and several studies support the validity of this criterion [50,51]. The internal consistency coefficients (Cronbach’s alpha) found in this study for the different dimensions were 0.79 (academic dimension), 0.77 (social dimension), 0.83 (emotional dimension), 0.79 (family dimension) and 0.69 (physical dimension). 

The Trait Meta-Mood Scale (TMMS-24; [52]) assesses perceived intrapersonal EI, i.e., each person’s knowledge of his or her own emotional states. Other authors performed different factor analyses before concluding that the items of this self-report are homogeneous and that the three subscales consistently measure the characteristics for which it was designed [53]; therefore, they are reliable and demonstrate construct validity. The internal consistency coefficients (Cronbach’s alpha) found in this study were 0.90 (attention to feelings), 0.90 (clarity of feelings) and 0.86 (repair of mood).

For the State-Trait Anxiety Inventory (STAI; [54]), the Spanish version of this test was used to assess anxiety, consisting of two scales of twenty items each, which measure two dimensions of anxiety, state and trait. According to [55], after making the Spanish adaptation in 1982, a reliability analysis was carried out using Cronbach’s alpha, obtaining 0.90 for trait anxiety (TA) and 0.94 for state anxiety (SA). 

To assess TA and SA in younger participants, we used the STAIC developed by [56,57] in California, as adapted to the Spanish population by [58]. This test has similar characteristics to the previous one, but the age range is between 8 and 15 years old.

### 2.3. Procedure and Characterisation of the Intervention

Firstly, we met with the school principals to inform them about the research, and informed consent was sought from the parents. All procedures performed conformed to the ethical standards of the University of Extremadura (Ref.:185//2020) and to the 1964 Helsinki declaration and its subsequent amendments or comparable ethical standards. All subjects gave informed consent for inclusion before participating in the study. The tutors of all the ESO, PCPI and DC courses were then contacted. A date and time for the tests was agreed with all of them. They were administered collectively in the classroom by the tutors and in the case of PCPI and DC students by the school guidance counsellors.

Clarifying the assessment of the students’ needs, as a starting point, we can say that in this case, no specific test was carried out beforehand, nor were the students’ needs assessed, but rather, the center’s team of counsellors, through observation, was the technique that allowed the appropriate information to be gathered from those subjects who voluntarily came to these professionals. In this sense, throughout the work experience in the academic center, the needs of the students were obtained, being very similar among them.

In short, the team of counsellors ensured that the intervention was adapted to the characteristics and needs of the subjects, thus allowing the intervention programme to be adjusted and carried out with an adequate sample, working with them on the different socioemotional processes described in this work.

Students were guaranteed anonymity and were asked to be always honest and cooperative. Likewise, we adapted to the dynamics of the educational center, which is why the sessions were 55 min long, coinciding with the usual duration of classes at the center.

In the characterisation of the intervention, all sessions contained the same outline and were organised in the same way. The structure can be observed in the example shown of one of the applied sessions. Each one of them presents a justification that allows the teacher to understand why it is necessary to carry out the activity; the objectives to be worked on with the activity; the development of the session, which has a beginning to motivate and awaken the student’s interest in carrying out the activity, a development, where the activity to be carried out is explained and an evaluation, which showed us whether the objectives of the session had been achieved or not and finally, the materials necessary to carry out the task. It should be noted that the application of the strategy followed is pedagogically the most used structure, so this type of organisation favours the approach and the implementation of the activities in a meaningful way.

In addition to the training carried out with the implementers of the activities, a manual was also produced so that the teachers could use it and implement it autonomously, without the need to resort to the main counsellor of the intervention programme. The handbook was fully detailed and was adjusted to the development time allocated to the specific tutoring hour. Of the different basic forms of intervention programmes, we opted for the application of this one using tutors as facilitators of the educational process to develop the personality of the students in a comprehensive manner.

Previously, a training session was held during which one of the guidance counsellors systematically trained the tutors in the implementation of the intervention programme, given that they were responsible for its execution in their tutoring hours. For this purpose, the guidance coordination time was used to advise and guide the tutors for each of the sessions. These meetings were held monthly and involved constant exchange of information to evaluate the implementation of the previous session and determine, in the teacher’s opinion, whether the objectives set had been met, what difficulties had been encountered and so on.

The person in charge of training the tutors has a degree in Psychopedagogy and has complementary and specialised training on the approach to the different socioemotional processes in the secondary school stage. It should be noted that teacher training depended on the affinity and the way of transmitting and knowing how to reach their students, as each one of them has different characteristics and personal resources that vary when it comes to tackling the activities for which they were trained.

The programme was structured on two levels, one for 1st and 2nd year ESO and the other for PDC, PCPI and 3rd and 4th year ESO. Each level comprised six sessions that ran from October to March, with one session per month. With regard to justifying the relevance of including different socio-emotional processes in the intervention programme, it was based on the need to respond to the demands of the students who came to the guidance department. Therefore, the team of counsellors were the ones who perceived the shortcomings and needs presented by the subjects, being the main reason why it was decided to work on and address the socioemotional processes described in this article. In order to ensure that the intervention programme was equally available to be applied and that it responded to the purpose of the research, the social, personal and cultural characteristics of the control group were similar to those of the experimental group. This ensured that both the pretest and the posttest were carried out appropriately in each case. Some of the characteristics that allowed us to select the sample were that the school was rural and not urban, that the school did not have excessive resources and that they were in localities with similar characteristics at the population level.

As for the intervention approach of the control group, it can be said that the subjects belonging to this group continued their normal daily routine, without emphasising those socioemotional processes that were implemented in the intervention programme. Although, it is true that during their routine, it is possible that different emotional aspects were involved in the activities carried out during the school day, such as self-esteem, emotional intelligence, assertiveness, empathy, among others. Therefore, these activities were not as focused or detailed as those specifically proposed by the experimental group, so that no intervention was made at any time on the aspects that the teachers had to work on with the participants in the control group.

The intervention programme was applied once a month for six months, with the intention of not manipulating the participation and intervention of the subjects and for it to be carried out in a natural environment that was not too distorted to reality. In other words, the aim was not to prioritise the programme applied, ignoring, and avoiding other important aspects that are dealt with during school hours, but rather to take into account the rest of the work and activities that form part of the day-to-day life of the educational centre.

In the same way, this periodisation allowed both the subjects in the experimental group and the implementers of the activities to familiarise themselves with the intervention programme, while always respecting the other needs of pupils of this age group. The aim was not to monopolise everything during the tutoring hour, but to address other relevant topics, such as health, eating habits, addictions, sexuality, etc., which are essential topics at these stages, and which should be worked on in the different academic years.

In short, the gradual incorporation of the programme makes it possible to integrate and coordinate the normal life of the center. This type of approach also contributes to having a greater depth and durability over time on the part of the participating subjects, as it would allow a connection and a link with the variables under study.

Set out below (Table 1 and Table 2) is the structure of the intervention programme, divided into blocks of content according to the objectives established.

Below is a type of example of the activities that make up the different sessions to clarify and enrich the understanding of the intervention program addressed (Table 3).

### 2.4. Statistical Analysis

To determine the nature of the data, the statistical programme SPSS 25 (Statistical Package for the Social Sciences, IBM Corp. Released 2012. IBM SPSS Statistics for Windows, version 25, IBM Corp., Armonk, NY, USA) was used. In accordance with the objective of this study and depending on the nature of the variables analyzed, descriptive and inferential data analyses were carried out. Student’s *t*-test was applied in the pretest to determine whether the groups were homogeneous or not. T-test for paired samples was also shown to investigate the differences in means between the two intervention times in the two groups. To evaluate the effect of the intervention program in each of the groups, the mean levels were compared using the repeated measures model. Tukey’s post hoc test was then performed when there was an interaction in the data. The confidence level considered was 95%.

## 3. Results

First, we analyzed whether there were differences in the pretest of self-concept, emotional intelligence, and anxiety in both groups. A Student’s *t*-test was used for this purpose. It was determined that, except for the social and family self-concept dimension, the rest of the variables included in the analysis showed that there were no differences between the control group and the experimental group (*p* > 0.05), so we can say that the groups were homogeneous at first.

The results of the descriptive and inferential analyses of the variables analysed in the study before (pretest) and after the intervention (posttest) are shown below.

Firstly, inferential analysis of the self-concept variable was carried out. Subsequently, and in order to check whether the intervention in the experimental group was effective in the self-concept variable, we compared the averages of both populations (Table 4). On this occasion, we used the paired sample *t*-test as the subjects were assessed in relation to two different conditions.

As can be observed (Table 4), there were no statistically significant differences between the pretest and posttest (*p* > 0.05) in the emotional dimension of self-concept for the experimental group. However, there were significant differences in the control group (*t* = 2.712; *p* = 0.007). The values obtained in this dimension show that the pretest of emotional self-concept is higher in both groups, and the difference between pretest and posttest in the control group was greater than in the experimental group. Regarding the effect size, the social, family, and academic dimensions had a small effect size, while in the case of the emotional dimension and physical dimension, it was non-existent.

In addition, following application of the programme in the experimental group, the social dimension of self-concept increased significantly (*t* = −2.299; *p* = 0.023). On the other hand, the control group also showed a difference (*t* = 2.701; *p* = 0.007) but in the opposite direction. That is, their scores were higher in the pre-test than in the post-test. In this case, it can be observed that in the control group, there is a decrease in the scores, while in the experimental group the same occurs in the control group. In the family dimension of self-concept, statistically significant differences were observed (*t* = −4.087; *p* = 0.00) but only in the case of the experimental group. In this dimension, students in the control group obtain a slight decrease in their scores, while in the experimental group, the resulting values are much higher in the posttest.

As for the academic dimension of self-concept, no statistically significant differences were found between the pretest and the posttest of the experimental group. However, significant differences were obtained for this dimension (*t* = 2.613; *p* = 0.010) in the case of the control group.

In relation to the physical dimension of self-concept, the data indicate (Table 4) that there were no statistically significant differences in either the experimental or the control group, but nor were there any statistically significant differences between the pretest and the posttest (*p* > 0.05).

To sum up, the programme was effective in the family and social dimension of self-concept, while in the other dimensions, the changes were not statistically significant. At the sample level, a decrease was detected in the levels of the academic and emotional dimension and a slight increase in the physical dimension. Therefore, the working hypothesis is partially confirmed.

Secondly, with regard to the emotional intelligence variable and focusing on emotion perception, understanding and regulation, in relation to the intervention programme, the results obtained were as follows (see Table 5).

As can be observed, there are no statistically significant differences (*p* > 0.05) for emotion perception between the pretest and the posttest in the experimental group, although the same does not occur with the control group (*t* = 3.336; *p* = 0.001). In other words, the difference in means of the control group is greater than that of the experimental group, but in both groups, there is a difference in values between the pretest and the posttest.

On the other hand, for the variable relating to emotion understanding, the values in the inferential analysis indicated an absence of significant differences between the pretest and posttest in the experimental group (*p* > 0.05). In contrast, the control group did show significant differences (*t* = 3.445; *p* = 0.001), showing, in both cases, that the values follow a downward trend.

As can be observed, the data indicate that for the emotion regulation variable, there were significant differences (*t* = 2.326; *p* = 0.021) between the pretest and the posttest of the control group, with none being found in the experimental group. The levels of both groups decrease in emotional regulation, with higher differences in the control group.

Thirdly, the results obtained for the trait anxiety (TA) and state anxiety (SA) variables are shown in Table 6.

Looking at the results, they indicate that there were no significant differences between the pretest and posttest of the CG and the EG for TA and SA (*p* > 0.05). With regard to the effect size, there was no effect on any of the variables under study among the groups analysed.

In Table 7, the results regarding self-concept show that, in the academic dimension, there is no interaction, i.e., the evolution from pretest to posttest is similar in both groups, so we cannot conclude that the intervention is distinguished in both groups. In this case, there is a decrease (F = 10.370; *p* = 0.001). The same occurs in the emotional dimension (F = 5.330; *p* = 0.02) and in the family dimension (F = 6.172; *p* = 0.01). The physical dimension does not show an interaction either, since the evolution in both cases is not significant, showing very similar values.

On the other hand, in the social dimension, there is a statistically significant interaction between both groups (F = 12.439; *p* < 0.001). The direction of the results is inverse, since the SG presents lower initial values in the pretest and improves in the posttest, while the opposite occurs in the CG. These differences are observed between the pretest of the SG and the pretest of the CG (*t* = −2.908; p_tukey_ = 0.02), and between the pretest of the CG and the posttest of the SG (*t* = 2.655; p_tukey_ = 0.04).

Focusing attention on emotional intelligence, the variable emotion perception, emotion understanding and emotion regulation in all cases, the two groups, in both the pretest and the posttest, are the same, with no evolution being observed. Therefore, we have a parallel model.

Finally, in the anxiety variable, we observe that there is a change in the means, but the temporal evolution shows non-significant values. We can also observe that the values of the CG are somewhat lower than those of the CG for AT and SA.

## 4. Discussion and Conclusions

The aim of this study was to evaluate the effectiveness of the implementation of a socio-emotional intervention programme; more specifically, the variables of the different dimensions of self-concept (emotional, social, family, academic and physical) and emotional intelligence (emotion perception, understanding and regulation). The results were not as was expected, given that there was no improvement in EI and the anxiety levels did not decrease. However, the social dimension of self-concept benefited, and significant changes were observed. The same did not occur in other interventions, such as [45,59], whose results for global self-concept and perceived social support demonstrated the effectiveness of the intervention programme they used for the improvement of self-concept and academic performance.

Regarding the influence of the intervention programme on the EI variable, studies such as [33,60] obtained significant improvements, mainly in relation to facilitation skills and socio-emotional development. Authors such as those in ref. [37] showed that an intervention programme carried out with adolescents between 13 and 16 years of age increased their EI, particularly between the pretest and the posttest for intrapersonal and interpersonal intelligence and mood. Previously, ref. [61] carried out an intervention programme aimed at adolescents aged 12 to 14 years to promote socio-emotional development and prevent violence. The results showed an improvement in violence prevention and EI with a significantly positive effect, contrary to the present case, which did not obtain better results in the EI construct.

A more recent study by [62] carried out an intervention programme to improve emotional skills, positivity, and empathy among adolescents. Following the intervention, the results indicated that the subjects’ EI improved, with a greater ability to perceive, manage, regulate, and understand their own emotions. In addition, adolescents who showed little ability to manage their own emotions also expressed unpleasant feelings.

On the contrary, the data obtained by [63] were similar to those obtained in our research. In their study, the aim was to improve self-esteem and reduce anxiety, although the most relevant results were obtained in relation to self-concept, and the same did not occur with the perception of stressful situations or the treatment of anxiety.

Other studies [64,65,66] on the prevention of anxiety in the adolescent population through an intervention programme found a decrease in anxiety symptoms and their negative effects and anxiety in general, results that are very close to those found in our research.

There is no unanimity when determining the effects of intervention programmes. In this sense, ref. [67] points out that the older the age of the participants, the more significant the emotional changes generated tend to be. Despite the results obtained by these authors, other studies have found the opposite [60].

For the moment, there is some evidence in favour of the susceptibility of educational intervention for certain dimensions of EI capacity and EI traits [46]. Although it is generally accepted that EI can be taught, the desired results are not always achieved following the application of specific programmes.

It is not enough to simply design and implement educational programmes that aim to develop EI; it is also necessary to assess the effectiveness of the interventions and gather empirical data regarding their degree of validity or simply to detect aspects of the interventions that could be improved [46]. Along the same lines, ref. [68] points out that the lack of systematisation and methodological rigour explain the difficulties in obtaining accurate and valid results in relation to the evaluation and effectiveness of emotional intervention programmes.

It is possible that differences in the appropriate effectiveness of interventions are a consequence of certain aspects, such as the objectives of the programme, the number of sessions or even the actors involved in the intervention. Although the findings in this paper extend the previous data on the effectiveness of emotional intervention programmes, certain limitations of our research should be mentioned.

One of the limitations is that no sessions were designed for the families as most of them live in farms, remote villages, and rural areas, and so their availability and participation was not guaranteed. We are convinced that socio-emotional education does not have a single target intervention context, the educational context, but must encompass other areas, such as the family and the community. Another limitation refers to the age range or developmental stage, since socio-emotional education is recommended throughout the life cycle. In this respect, ref. [60] points out that for an emotional education programme to be relevant, it is necessary to involve parents and school staff, given that the family and school context can interfere with adolescents’ tasks and activities.

The intervention programme designed and applied by our group was focused on developing and strengthening emotional skills in general, and the teachers involved found it very interesting and were available at all times. However, we have to recognise that this activity increased their workloads, as it was an extraordinary job.

Given the characteristics of the sample, the number and content of sessions should be reviewed, as we wanted to make the sessions more general, integrated, and versatile rather than being specific to the constructs analysed. This may explain the lack of improvement in most of the variables studied. It is likely that with a more extensive programme focusing on activities and objectives to improve a single construct, the results would have been better for the variables considered.

We intend to continue implementing socio-emotional intervention programmes with larger and more varied samples, reviewing activities, and involving families and other stakeholders in the educational community, such as social workers, management teams, etc. Another possibility is to increase the number of sessions and focus more directly on the proposed objectives. Similarly, more exhaustive assessment of both academic performance (considering the grades of the last two years in each of the subjects) and physical self-concept (with a more specific test and asking questions about physical exercise, types of sport, how much time they devote to it each week, etc.) is another possible option. This would provide more relevant data regarding the dimensions of self-concept for the students concerned.

Finally, in order to assess which type of intervention is more effective, several programmes could be applied for a different population sample and a comparative study could be carried out to evaluate the results in order to decide which of them is the most effective in terms of the objectives to be achieved.

Therefore, despite the limitations and taking into account certain suggestions to rectify and improve future intervention proposals, we are confident that the findings of this paper are innovative, relevant and interesting for guidance departments when implementing intervention programmes to train and develop emotional skills through the Tutorial Action Plans. We believe that such measures aimed at optimising emotional, affective and social skills will not only protect adolescent students from academic and social imbalances, but they will also improve their comprehensive development, which is ultimately one of the most necessary and at the same time most difficult goals to achieve.

## Figures and Tables

**Table 1 ijerph-19-08153-t001:** Structure of the intervention programme for 1ST and 2ND YR ESO.

				1ST and 2ND YR ESO
Session	Strategy	Block	Title	Objectives
1 October	Sessions with a basic outline (initial phase, development, and assessment). The methodology applied is active, participative, and adapted to the needs of the students, working on the variables at an integral level.	SELF-CONCEPT AND SELF-ESTEEM	“AMULETS”	Promote positive self-image. Make students aware of their personal skills that justify and explain many of their successes.
2 November		ASSERTIVENESS	“EXCUSE ME BUT...”	Identify the types of behaviour that can be useful when dealing with different situations. Learn ways to improve relationships with others. Reflect on one’s own behaviour towards others and make suggestions for improvement.
3 December		EMPATHY	“PUTTING MYSELF IN YOUR SHOES”	Encourage empathy (putting yourself in other people’s shoes). Help to understand different perspectives of a problem or conflict.
4 January		SELF-CONTROL	“WHAT A TEMPTATION!”	Appreciate the importance of developing the capacity for self-control. Analyse and reflect on the different ways of controlling oneself in different situations.
5 February		DECISION-MAKING	“NOW … WHAT DO I DO?”	Acquire a general understanding of what it means to ‘make decisions’ and recognise the importance of learning to make decisions. Identify the correct way to make a decision in simulated situations. Reflect on the importance of making decisions and taking responsibility for one’s actions.
6 March		RESPONDING TO CRITICISM	“EXPRESSING AND ACCEPTING CRITICISM”	Reflect on what criticism is. Understand the importance of knowing how to respond correctly to criticism in order to avoid anger, conflicts or fights. Put into practice the lessons learnt in previous sessions on assertive, passive, and aggressive styles.

Source: prepared by the author.

**Table 2 ijerph-19-08153-t002:** Structure of the intervention programme for 3RD and 4TH YR ESO, DC and PCPI2.

				3RD and 4TH YR ESO, DC and PCPI2
Session	Strategy	Block	Title	Objectives
1 October	Sessions with a basic outline (initial phase, development, and assessment). The methodology applied is active, participative, and adapted to the needs of the students, working on the variables at an integral level.	SELF-CONCEPT AND SELF-ESTEEM	“TODAY I FEEL GOOD”	Encourage a critical sense and sense of self-assertion. Accepting ourselves as we are. Being able to reframe negative thoughts into positive messages.
2 November		ASSERTIVENESS	“YOU RESPECT ME, I RESPECT YOU”	Knowledge of the types of behaviour that can be useful when dealing with different situations. Reflect on the advantages and disadvantages of each one. Transfer or generalise the knowledge acquired in different real-life situations.
3 December		EMPATHY	“PUTTING MYSELF IN YOUR SHOES”	Learn to identify other people’s feelings. Help to understand different perspectives of a problem or conflict.
4 January		SELF-CONTROL	“I DON’T WANT ANY BAD VIBES”	Appreciate the importance of being able to control oneself. Put into practice different techniques for self-control in different situations.
5 February		DECISION-MAKING	“EVERYTHING HAS ITS DRAWBACKS”	Reflect on one’s own personal decisions. Promote positive thoughts that help to hold a position more firmly.
6 March		RESPONDING TO CRITICISM	“FORMULATING AND RESPONDING TO CRITICISM”	Reflect on what constructive criticism is. Learn certain skills to formulate and respond to criticism. Put into practice the lessons learnt in previous sessions on assertive, passive, and aggressive styles.

Source: prepared by the author.

**Table 3 ijerph-19-08153-t003:** Example of activity type of block self-control.

Session 4.2	Title: “I DON’T WANT ANY BAD VIBES”	
Classes: 3RD and 4TH YR ESO, DC and PCPI2	Block: Self-Control	Duration: 55 min
**Justification**		
Self-control is not an easy subject to pass and even less so in adolescence. In the struggle with oneself to “not jump”, all possible strategies, both internal and external, must be employed in order to deal with situations successfully and not succumb to the temptation of impulsivity as a way of solving a problem. The following activity prepares them to consider the advantages and disadvantages of allowing themselves to be carried away by the moment or by provocation, leading them to reflect on the concept of self-control and the many ways of exercising it.		
**Objectives**		
- Evaluate the importance of being able to control oneself. - Put into practice different ways of control in different situations.		
**Development of the session**		
***Start*** The tutor starts the session: “Today we are going to reflect on the importance of being able to control ourselves in different situations and to be able to assess the consequences of our behaviour. Sometimes, it is very difficult for us to hold back the urge to do something, to say certain things... but if we look closely, we will see that it is very advantageous to stop for a second and think about what we are going to do and thus assess whether it is the best thing for us. Spending a few seconds on this already makes us behave in a self-controlled way”.		
***Development*** VIEWING THE SEQUENCE The activity is introduced with the following commentary: “Next, we are going to watch a sequence in which Raul, the main character, has to decide whether or not to respond to a provocation. I would like you to take a good look at his behaviour so that we can comment on it later”.		
The audiovisual is projected. After watching the sequence, a small debate is opened in which the group’s opinions about Raúl’s attitude, that of his friends, the possible help he receives, the impediments he encounters, etc. are explored, and finally, an attempt is made to describe what they understand by self-control.		
The discussion should also focus on the following aspects: - How they think one should respond to a provocation and make them understand that, deep down, to respond to a provocation is a form of obedience (something that, in essence, any adolescent rejects outright). - How they believe that substances, in this case alcohol, influence situations such as these that occur regularly.		
The following comment is made: “Self-control is, in short, the ability to be able and know how to stop, not letting ourselves be driven by impulses, but by our heads”.		
*EXERCISE “PREPARING THE TOOLBOX”.* The activity is continued by making the following comment: “Despite what it may seem to us, self-control does not come “as standard” in any of us. Rather, those who tend to show this ability often have their “tricks” that allow them to be guided less by impulses and more by thinking before they act. In this section, we are going to prepare our own “toolbox”, i.e., everything we would need to be able to do our job (in this case, to control ourselves in the face of a provocation) and come out of it having known how to control ourselves”. The class is divided into teams of 2–3 members, whose mission will be to identify the weapons and strategies that Raúl has put in place to avoid “getting into a rage”, making a distinction between those that come “from within” (i.e., that he himself puts in place as part of his personal resources), and those that come from outside (i.e., that are provided by the people around him or by the situation itself). This distinction makes sense, especially considering that the concept of self-control that we want to emphasise is not that of an all-powerful “superman”, but that of someone who intelligently uses all the resources at his disposal in order not to get carried away by a situation that could be detrimental to him. When the different groups have discussed and pointed out the different ways of self-control that Raúl has used, a small group discussion is held, which in turn leads to the next phase of the activity. The following question is asked: do you control yourself? The next phase goes one step further. The different teams have to complete the list of resources made in the second phase with others of their own or that they can think of, and that they can use in situations where they have to control themselves. To make the task easier, they can think of examples of situations in which they controlled themselves (to think about what they used at those times) or think of times when they did not manage to do so and consider what they would have needed to achieve it. Finally, a single list will be made of all the proposals made by the teams. Emphasis will be placed on the fact that there are many ways of self-control; therefore, acting on impulse is not the best option, nor is it as inevitable as we sometimes think or would have us believe.		
***Assesment and concluding remark*** “We have seen that it is very important to be able to stop and think about what we feel like doing, and especially what might happen later, rather than acting impulsively. Often our impulses are very strong, so it is important to have a certain level of self-control: in what we think, what we feel and what we do”.		
** *Materials* **		
− Sequence is as follows: passing of bad vibes Session extracted from the following resource: Build your world. Prevent to live. Retrieved from http://www.construyetumundo.org (accessed on 17 January 2022)		

Source: prepared by the author.

**Table 4 ijerph-19-08153-t004:** Inferential analysis of the self-concept of the experimental group (EG) and the control group (CG) pretest-posttest.

	Emotional Dimension		Social Dimension		Family Dimension		Academic Dimension		Physical Dimension	
	E.G.	C.G.	E.G.	C.G.	E.G.	C.G.	E.G.	C.G.	E.G.	C.G.
**M**	4.099	6.205	−6.125	6.643	−10.047	1.638	2.719	6.929	−0.094	−2.624
**SD**	30.788	33.150	36.913	35.644	34.060	38.552	39.386	38.431	33.805	30.009
**T**	1.845	2.712	−2.299	2.701	−4.087	0.616	0.956	2.613	−0.038	−1.267
**DF**	191	209	191	209	191	209	191	209	191	209
***p*-value**	0.067	0.007	0.023	0.007	0.000	0.539	0.340	0.010	0.969	0.207
** *D* **	0.066		0.352		0.32		0.108		−0.079	
**Confidence interval**	−0.13	0.261	0.155	0.549	0.123	0.517	−0.088	0.304	−0.275	0.116

Source: prepared by the author. M: mean; SD: standard deviation.

**Table 5 ijerph-19-08153-t005:** Inferential analysis of the emotional intelligence (EI) of the EG and the CG in the pretest-posttest.

	Emotion Perception		Emotion Understanding		Emotion Regulation	
	E.G.	C.G.	E.G.	C.G.	E.G.	C.G.
**M**	0.036	0.181	0.42	0.195	0.115	0.133
**SD**	0.888	0.786	0.867	0.821	0.908	0.831
**T**	0.569	3.336	0.666	3.445	1.749	2.326
**DF**	0.191	209	191	209	191	209
***p*-value**	0.570	0.001	0.506	0.001	0.082	0.021
** *D* **	0.173		−0.267		0.021	
**Confidence interval**	−0.023	−0.369	−0.463	−0.07	−0.175	0.216

Source: prepared by the author. M: mean; SD: standard deviation.

**Table 6 ijerph-19-08153-t006:** Inferential analysis of anxiety in the EG and the CG in the pretest-posttest.

	TA		SA	
	E.G.	C.G.	E.G.	C.G.
**M**	−0.984	−5.438	−0.78	1.243
**SD**	44.770	40.394	41.116	40.536
**T**	−0.305	−1.951	−0.026	0.444
**DF**	191	209	191	209
** *p* ** **-value**	0.761	0.052	0.979	0.657
** *D* **	−0.105		0.05	
**Confidence interval**	−0.301	0.091	−0.146	0.245

Source: prepared by the author. M: mean; SD: standard deviation.

**Table 7 ijerph-19-08153-t007:** Results of the repeated measures model for each variable of self-concept, emotional intelligence, and anxiety.

Variables		C.G.	E.G.	F	*p*	η^2^
**Self-concept**	Emotional Dimension	Pretest	57.910 ± 1.972	5.330	0.02 *	0.005
		Posttest	56.271 ± 1.941			
	Social Dimension	Pretest	62.362 ± 1.898	0.020	0.88	0.000
		Posttest	55.719 ± 2.024			
	Family Dimension	Pretest	59.810 ± 1.972	6.172	0.01 **	0.007
		Posttest	52.881 ± 2.081			
	Academic Dimension	Pretest	60.014 ± 1.740	10.370	0.00 ***	0.100
		Posttest	53.810 + 1.847			
	Physical Dimension	Pretest	68.900 ± 1.677	0.729	0.39	0.001
		Posttest	71.524 ± 1.721			
**Emotional Intelligence**	Emotion Perception	Pretest	1.733 ± 0.044	6.776	0.00 ***	0.007
		Posttest	1.552 ± 0.042			
	Emotion Understanding	Pretest	1.738 ± 0.43	7.910	0.00 ***	0.009
		Posttest	1.543 ± 0.041			
	Emotion Regulation	Pretest	1.971 ± 0.044	8.172	0.00 ***	0.009
		Posttest	1.834 ± 0.046			
**Anxiety**	Trait Anxiety	Pretest	31.062 ± 2.084	2.286	0.13	0.003
		Posttest	36.500 ± 2.084			
	State Anxiety	Pretest	42.871 ± 2.101	0.082	0.77	0.000
		Posttest	41.629 ± 2.049			

* *p* < 0.05; ** *p* < 0.01; *** *p* < 0.001.

## Data Availability

The data presented in this study are available upon request from the corresponding author.

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
