# Peer review of "Efficacy of a Psycho-Educational and Socio-Emotional Intervention Programme for Adolescents"

_ijerph, 2022, doi:10.3390/ijerph19138153_

Round 1
Reviewer 1 Report
Comments and suggestions for authors are attached below

Reviewer 2 Report
Thank you very much for giving me the opportunity to review this paper. I find it impeccable from a methodological point of view. Easy to read, great bibliographical support and meets all the requirements to be published. My sincere congratulations to the authors.
Reviewer 3 Report
Dear authors:
The manuscript is of interest/relevant.
However, I think the authors need to improve it.
My suggestions/recommendations:
- on page 3 (line 139): correct "objectives" to "objective";
- strengthen the "introduction" with more current references ( in the period 2019- 2021 only 5 references are presented);
-need to improve table 1; I suggest: split the table in two (separate 1ST & 2ND from 3RD & 4TH); introduce lines separating each session; introduce a column referring to the strategy followed.
- in relation to sub-section 2.3, there is a notorious lack of characterization of the intervention. For example: they need to describe the strategy followed (and why this strategy...) in the sessions; they need to describe the duration of each session.
Rev
